# Biological Containment for African Swine Fever (ASF) Laboratories and Animal Facilities: The Italian Challenge in Bridging the Present Regulatory Gap and Enhancing Biosafety and Biosecurity Measures

**DOI:** 10.3390/ani14030454

**Published:** 2024-01-30

**Authors:** Silvia Pavone, Carmen Iscaro, Monica Giammarioli, Maria Serena Beato, Cecilia Righi, Stefano Petrini, Silva Costarelli, Francesco Feliziani

**Affiliations:** 1National Reference Laboratory for Pestivirus and Asfivirus, Istituto Zooprofilattico Sperimentale dell’Umbria e delle Marche “Togo Rosati” (IZSUM), Via G. Salvemini 1, 06126 Perugia, Italy; c.iscaro@izsum.it (C.I.); m.giammarioli@izsum.it (M.G.); ms.beato@izsum.it (M.S.B.); c.righi@izsum.it (C.R.); s.petrini@izsum.it (S.P.); f.feliziani@izsum.it (F.F.); 2Animal Health Diagnostic Laboratory, Istituto Zooprofilattico Sperimentale dell’Umbria e delle Marche “Togo Rosati” (IZSUM), Via G. Salvemini 1, 06126 Perugia, Italy; s.costarelli@izsum.it

**Keywords:** African Swine Fever, animal facility, biological containment measures, biorisk management, biosafety, biosecurity, internal audit, laboratory, risk assessment

## Abstract

**Simple Summary:**

The African Swine Fever Virus (ASFV) is a DNA virus belonging to the *Asfivirus* genus and Asfarviridae family. In recent years, ASF has grown to be a significant concern for the pig industry worldwide. The ever-growing handling of ASFV pathological material in laboratories, for either diagnostic or research activities, requires particular caution to avoid accidental virus release from laboratories with consequent detrimental economic and environmental effects. This paper aims to describe the Italian National Reference Laboratory (NRL) efforts to develop a biological containment tool for ASF laboratories and animal facilities. The easy-to-apply recommendations focusing on the risk assessment system and internal audits used for enhancing Italian biosafety and biosecurity measures are reported.

**Abstract:**

The African Swine Fever Virus (ASFV) is a DNA virus of the Asfarviridae family, *Asfivirus* genus. It is responsible for massive losses in pig populations and drastic direct and indirect economic impacts. The ever-growing handling of ASFV pathological material in laboratories, necessary for either diagnostic or research activities, requires particular attention to avoid accidental virus release from laboratories and its detrimental economic and environmental effects. Recently, the Commission Delegated Regulation (EU) 2020/689 of 17 December 2019 repealed the Commission Decision of 26 May 2003 reporting an ASF diagnostic manual (2003/422/EC) with the minimum and supplementary requirements for ASF laboratories. This decision generated a regulatory gap that has not been addressed yet. This paper aims to describe the Italian National Reference Laboratory (NRL) efforts to develop an effective and reliable biological containment tool for ASF laboratories and animal facilities. The tool consists of comprehensive and harmonized structural and procedural requirements for ASF laboratories and animal facilities that have been developed based on both current and repealed legislation, further entailing a risk assessment and internal audit as indispensable tools to design, adjust, and improve biological containment measures.

## 1. Introduction

Veterinary laboratories and animal facilities routinely handle biological material that may contain infectious agents and toxins that may pose risks to animal and/or public health as well as causing adverse economic effects [1]. Their occurrence is framed as a biorisk. Biorisk management (BRM) is a systematic way of addressing and controlling risks encountered in laboratories, a process that is necessary to protect laboratory workers, animals, and the human population from inadvertent exposures and infections whilst also protecting the environment from accidental or intentional release and subsequent spread of biological agents and their toxins from laboratories [1]. BRM includes a range of documented processes and physical infrastructure which are developed to ensure that both biosafety and biosecurity risks are adequately controlled. According to the World Health Organization (WHO), biosafety is defined as the safe working practices associated with the handling of biological materials, particularly infectious agents. It addresses containment principles, technologies and practices that are implemented to prevent the unintentional exposure to pathogens and toxins, or their accidental release [2]. Biosafety involves administrative controls, standard operating procedures (SOPs), personal protective equipment (PPE), and engineering control [3]. On the other hand, biosecurity focuses on personal and institutional security measurements to protect bioagents from unauthorized access, misuse, thievery, and intentional release [4].

The biosafety and biosecurity of laboratories and animal facilities are based on risk assessment and containment. Risk assessment is the process that allows for the identification of the hazard and its source, establishes the probability of intentional or accidental spreading, and identifies the probability of exposure to contagion (animals or humans) or contamination (environment) [5]. Moreover, it allows for the severity of the epidemiological and economic consequences to be established, as well as for the practices that require implementation to prevent laboratory-associated infections (LAI) to be identified. The primary risk criteria used to define the four ascending levels of containment, referred to as Biosafety Levels 1 to 4 (BSL-1, BSL-2, BSL-3, BSL-4), are: 1. infectivity, 2. severity of disease, 3. transmissibility, and 4. nature of the work being conducted. Another important risk factor for the agents that cause moderate to severe disease is the agent’s origin, i.e., whether it is indigenous or exotic. Each level of containment describes the microbiological practices, safety equipment, and facility safeguards for the corresponding level of risk associated with the handling of an agent. The facility safeguards associated with BSL-1 through BSL-4 help protect the non-laboratory occupants of the facility, the public health, and the environment [6]. On the other hand, the fundamentals of containment include microbiological practices, safety equipment, and facility safeguards (PPE) that protect laboratory workers, the environment, and the people from exposure to infectious microorganisms that are handled and stored in the laboratory and/or animal facilities [7]. Worldwide, the accepted classification systems adopted by most international organizations and countries for the different laboratory protection levels are BSL-1, BSL-2, BSL-3, and BSL-4 for facilities designed to handle suitable pathogens in vitro; similarly, ABSL-1, ABSL-2, ABSL-3, and ABSL-4 (animal biosafety laboratory) are used for facilities designed to handle suitable pathogens for in vivo experiments [8].

The African Swine Fever virus (ASFV) is a DNA virus of the Asfarviridae family, *Asfivirus* genus [9]. ASFV is the sole member of its family and the only known DNA arbovirus [10]. The disease does not pose a risk to human health, but is characterized by high morbidity and lethality rates in swines [9]. ASF occurs through transmission cycles involving domestic pigs, wild boars, wild African suids, and soft ticks (*Ornithodorus* spp.) [11]. The ASFV can spread by direct contact between sick and healthy animals as well as indirectly by feeding on garbage containing infected meat (ASFV can remain infectious for 3–6 months in uncooked pork products), mechanic (premises, vehicles, implements, clothes, etc.), or biological vectors (ticks) [12,13,14]. Responsible for massive losses in pig populations and drastic economic consequences, ASF has become a major concern for the swine industry in recent years [15]. ASF is currently affecting several regions around the world and, with no effective vaccine, the disease is not only affecting animal health and welfare, but also, in certain areas, biodiversity and farmers’ livelihoods [16].

Based on a risk assessment for ASFV, the Centers for Disease Control and Prevention National Institutes of Health (CDC) considers ASFV an agent that needs biosafety level 3 (BSL-3) in laboratories and biosafety level 3 (ABSL-3) in animal facilities [6] where swines are used in vaccine efficacy and pathogenicity studies, as well as in the production of reference material.

Recently, the Commission Delegated Regulation (EU) 2020/689 of 17 December 2019 [17] repealed the Commission Decision of 26 May 2003 reporting an African Swine Fever diagnostic manual (2003/422/EC) [18] and the minimum/supplementary safety requirements for ASFV BSL-3 laboratories and animal facilities. This decision created a regulatory gap that has not been addressed yet. Currently, Directive 2000/54/EC of the European Parliament and of the Council of 18 September 2000 on the protection of workers from risks related to exposure to biological agents at work [19] is the only legally recognized measure with respect to containment levels. However, these provisions are specific to the protection of laboratory workers rather than public, animal, and environmental health. 

In the current epidemiological scenario, with ongoing ASF outbreaks in several countries across Asia, the Caribbean, Europe, and the Pacific [20,21,22], it is necessary to promote research on this disease and develop biosecurity measures that are clear, applicable, and cost-effective. Since vaccine development remains a major gap in ASF control and eradication, in vivo and in vitro experiments are essential to increase the knowledge of this disease [23]. On the other hand, the handling of pathogens that pose a high risk to animal health requires strict and specific guidelines recognized by the scientific community.

This paper aims to describe the Italian National Reference Laboratory (NRL) efforts to develop an effective and reliable biological containment tool for ASF laboratories and animal facilities. The tool consists of comprehensive and harmonized structural and managerial requirements for ASF laboratories and animal facilities, further entailing a risk assessment and internal audit system which represents an indispensable tool to design, adjust, and improve biological containment measures. The easy-to-apply structural and managerial requirements were developed considering both current and repealed legislation, the broad biosafety manuals and guidelines published by national and international agencies, and the long-time experience of the Italian NRL with regard to ASF.

## 2. Materials and Methods

The biological containment tool adopted by the Italian NRL for ASF was developed based on the following three main points: 1. risk assessment, 2. the easy-to-apply requirements for ASF laboratories and animal facilities based on both current and repealed legislation, the standard international guidelines as a reference framework, and the long-time experience of the Italian NRL, and 3. internal audit program.

### 2.1. The Italian Risk Assessment Guidelines for Accidental Release of ASFV Infected Material from the BSL-3 Containment Area

The developed risk assessment was based on the flowchart described in the WOAH Terrestrial Manual, Chapter 1.1.4. [24], suitably adapted to the scope. The present risk assessment used a qualitative system. For each source of an identified hazard, a judgment was expressed considering the likelihood of the risk-related event occurring and the severity of its consequences. For a better definition of the risk levels, a severity scale with five risk levels (“low”, “acceptable”, “medium”, “remarkable”, and “severe”) was defined. Low and acceptable levels were ascribed to the activities or areas identified as sources of hazards for which it is not necessary to implement targeted actions aimed at reducing the risk. Conversely, the risk attributed to the activities or areas identified as sources of hazards for which it is essential to implement targeted actions to reduce the risk was defined as medium, remarkable, and severe.

The risk level was based on the combination of the two variables, as reported in Figure 1: (1) the assessment of the likelihood of the risk-related event occurring and (2) the severity of the consequences engendered by the occurrence of the hazard. The first variable was defined by three increasing levels of probability (improbable, possible, and probable), the second variable was defined on a scale of three increasing levels (mild, medium, and marked).

The risk assessment must be completed prior to any laboratory activities taking place, as well as each time that new procedures and/or risks are identified or changes to the facility, equipment, or agents occur. The biosafety officer conducts the assessment annually, or whenever deemed appropriate, and must identify risk factors and potential sources of contamination that are linked to the identified hazard(s). This activity was carried out jointly with the laboratory and animal facility staff. When implementing the requirements for ASF laboratories and animal facilities, a specific risk assessment was carried out—following the scheme in Figure 1—for each source of hazards identified. The risk assessment, and, therefore, the identification of any preventive measures, was implemented to control and reduce the risk to an acceptable level. When risk mitigation was not possible, the biosafety officer informed the facility director about the identified hazard, as the director is able to order a suspension of work activities related to the process associated with an unacceptable risk factor. Here, we report the list of the eleven identified hazards for which a risk assessment was performed over time in the BSL-3 containment area of the Italian NRL for ASF.

Management, storage, and handling of samples contaminated or potentially contaminated with ASFV related to diagnostic and research activities.Infectious or potentially infectious waste (solid and liquid) produced within the BSL-3 containment area.Personnel involved in institutional activities or having access to the BSL-3 containment area.Material (documentation, equipment, tools, etc.) leaving the BSL-3 structure.Contamination resulting from experimental infections conducted in animal housing facilities.Intrusion by unauthorized personnel and possible theft of infected material.Spillage of contaminated material from the sewage system that leads the wastewater to the sterilization system.Entry of personnel involved in research activities using experimental animals housed in the ABSL-3.Interruption of services essential to the containment measures (electricity, water);Non-epidemic emergencies.Epidemic emergencies.

### 2.2. Requirements for ASF Laboratories and Animal Facility

The Directive 2000/54/EC of the European Parliament and of the Council of 18 September 2000 on the minimum health and safety requirements regarding the exposure of workers to the risks arising from physical/chemical/biological agents [19], as well as the repealed Commission Decision of 26 May 2003 [18] describing the ASF diagnostic manual (2003/422/EC) and the minimum safety requirements for ASF laboratories, were the regulations considered to develop the new ASF biosafety and biosecurity tool.

To provide updated, harmonized, and more comprehensive specific guidelines, the provisions in both legislations were reported in the present recommendations, avoiding redundancy to seamlessly integrate both guidelines and other indications extrapolated from the WHO Laboratory Biosafety Manual [25] and Biosafety in Microbiological and Biomedical Laboratories [6] that were deemed relevant. The scheme followed integrated the sections considered in the ASF diagnostic manual (2000/54/EC) (“General environment”, Laboratory clothing”, “Control of personnel”, “Equipment” for the diagnostic laboratories, and “General environment”, Laboratory clothing”, “Control of personnel”, “Equipment” and “Animals” for the experimental animal rooms) with newly added entries (“Specific disinfection procedures” and “Handling procedures of infected material” for the diagnostic laboratories, and “Specific disinfection procedures” for the experimental animal rooms) necessary to better clarify every aspect of the biosafety for ASF laboratories and animal facilities. 

The present requirements did not include general standard practices and procedures, basic training, and basic PPE necessary to ensure the biosafety of laboratories which was not specific to ASF. Therefore, some indications such as the ban on eating, drinking, smoking, contact lenses handling, cosmetics application, and storage of food for human consumption were not reported, unlike other biosafety manuals.

### 2.3. Internal Audit Activity of the NRL for ASF

The internal audit activity was also developed to ensure the proper functioning of the biological containment for ASF laboratories and animal facilities to avoid, in turn, accidental or intentional release of ASFV infected material outside the BSL-3 containment area. The checklists were designed based on the Veterinary Public Health Department of the Istituto Superiore di Sanità (ISS) biosecurity audit system and suitably adapted to our aim. Specific checklists for the general environment, policies, responsibilities, and SOP, personnel, and ASFV repositories were used. Here, we report the issues considered in the Italian NRL internal audit activity in relation to all the above mentioned aspects.

-Biosafety management system requirements, including risk management, analysis of accidents and injuries, monitoring and control of activities and emergency management, and change in management.-Infrastructure management that includes general issues (such as availability of testing documents on site and commissioning of the system, suitability of laboratory surfaces and general environment, etc.), availability of procedures on commissioning, disposal, and release of materials, access to the working area, and management of premises and equipment.-Storage and handling of ASFV, including traceability of the biological agents, SOP, inactivation of ASFV, waste management, clothing and PPE, transport, and shipment of ASFV.-Security management to minimize the risk of unlawful removal of ASFV from the facility.

## 3. Results

Herein we report: 1. the easy-to-apply requirements for ASF laboratories and animal facilities developed by the Italian NRL, 2. the results of the risk assessment applied to our ASF containment area, and 3. the results of the internal audit program performed over time in our laboratories.

### 3.1. Application of Risk Assessment Guidelines for Accidental Release of Infected Material by ASFV Outside the BSL-3 Containment Area

Here we report the risk assessment for ASFV performed at the start of the activity in the Italian NRL BSL-3 containment area, having identified eleven hazards: Management, storage, and handling of samples contaminated or potentially contaminated with ASFV related to institutional and research activities. All these activities represent a high risk source of environmental contamination outside the BSL-3 facility if the containment measures are not properly applied.Infectious or potentially infectious waste (solid and liquid) produced within the BSL-3 containment area.Personnel involved in diagnostic and research activities or having access to the BSL-3 containment area. The personnel operating in the BSL-3 containment area or any person entering even occasionally the facility is to be considered contaminated or potentially contaminated and, therefore, a possible carrier of the virus outside the facility, with the potential for infection of susceptible animals.Material (documentation, equipment, tools, etc.) leaving the BSL-3 infrastructure. All material leaving the BSL-3 containment area is to be considered potentially contaminated with ASFV and, therefore, a potential source of high risk environmental contamination.Contamination resulting from in vivo experimental studies conducted in the animal facility. The in vivo experimental studies with ASFV in the animal facilities represent a significant risk of environmental contamination, as the animals may become high virus shedders.Intrusion by unauthorized personnel and possible theft of infected material. The theft and intentional release in the environment of ASFV or contaminated material for bioterrorism purposes cannot be excluded.Spillage of contaminated material from the sewage system leading the wastewater to the sterilization system.Entry of personnel involved in research activities using experimental animals housed in the ABSL-3. The research activity requires entry to and exit from the animal facility of personnel who is to be considered potentially contaminated with ASFV and, therefore, a potential source of environmental contamination.Interruption of services essential to the containment measures (electricity and water).Non-epidemic emergencies. The most common non-epidemic emergencies are accidental spillage of material, fire, and earthquakes. In particular, the Italian NRL for ASF is located in Perugia, a town where severe earthquakes may occur.Epidemic emergencies. During an epidemic emergency, increasing analytical activities, flows, handling of samples, etc. may compromise the maintenance of biosafety measures in the BSL-3 premises.

The risk level associated with each hazard, except for hazard 7, was considered remarkable based on the developed diagram (Figure 1), as the possibility of environmental contamination results from the intersection of two events defined as “possible” (probability of the event) and “marked” (severity of the consequences). To reduce the risk to an acceptable level, specific procedures and measures were implemented.

More specifically, the following SOPs were identified and outlined:(a)Procedure reporting the provisions for biosafety management, storage, and handling of biological material infected or potentially infected with ASFV. This procedure was linked to a further practice reporting the provisions for dangerous goods transport by road, i.e., ADR (European Agreement concerning the International Carriage of Dangerous Goods by Road), rail, i.e., RID (Agreements Concerning the International Carriage of Dangerous Goods by Rail), air, i.e., ICAO (International Civil Aviation Organization), and sea, i.e., IMDG (International Maritime Dangerous Goodsk).(b)Specific waste management procedure reporting the requirements for management, decontamination, and disposal of waste infected or potentially infected with ASFV.(c)Procedure regulating access to/exit from the BSL-3 containment area.(d)SOPs on personnel behavior within the containment area.(e)SOPs on cleaning and disinfection of environments, surfaces, equipment, and tools.(f)SOPs on cleaning and disinfection of the animal facility.(g)SOPs on waste decontamination (solid and liquid effluents).(h)SOPs on the proper use and maintenance of the incinerator for carcasses disposal; SOPs on management of non-epidemic emergencies.(i)SOPs on the management of epidemic emergencies. A Crisis Unit designated to strategic choices and a special Emergency Team consisting of technical personnel adequately trained were identified.

Moreover, the following preventive measures were applied to:-Prevent unauthorized access to the BSL-3 area. A video surveillance system was installed to constantly monitor the perimeter outside the BSL-3 facility.-Ensure the supply of electrical and water services. Therefore, an electric generator was installed in the BSL-3 facility. When an interruption to the electricity occurs, the electric generator guarantees prolonged autonomy. Similarly, an adequate water supply was set up to guarantee occurrence of essential operations in support of the disinfection procedures.

Regarding point 7, the risk level was considered acceptable because the risk associated with occurrence of environmental contamination results from the intersection of two events defined as “improbable” (probability of the event) and “medium” (severity of the consequences). Indeed, although the spillage of contaminated material from the sewage system cannot be excluded, the system is underground and, therefore, a surface soil contamination is unlikely. Moreover, the BSL-3 containment area is located far from groundwater; therefore, any spillage of contaminated fluid effluents is likely to remain confined within the small underground area, thus posing no significant risk to suids. To further reduce the risk, periodic checks (or checks prompted by suspicion) of the tightness of the sewage system were envisaged.

### 3.2. Requirements for BSL-3 Containment Area for ASFV

The BSL-3 containment area can include an area for the laboratory and one for the animal facility, where the animals are housed during in vivo experimental trials. Within the facility where ASFV is handled, minimum biosecurity and biosafety requirements must be respected, according to the activity carried out and the area in which it takes place.

Minimum requirements for ASFV laboratories must be fulfilled when analyses using inactivated antigens are conducted, whereas maximum requirements must be fulfilled in any laboratory where the ASFV is amplified by in vitro replication into cell cultures.

The developed and adopted recommendations by the Italian NRL included specific requirements for ASF laboratories and animal facilities. Such requirements included those already listed in the repealed Commission Decision 2003/422/EC [18] which were related to the following sections: “General environment”, “Laboratory clothing”, “Control of personnel”, “Equipment” for diagnostic laboratories, and “General environment”, “Laboratory clothing”, “Control of personnel”, “Equipment” and “Animals” for the experimental animal rooms with the addition of new ones (i.e., “Specific disinfection procedures” and “Handling procedures with respect to infected material” for the diagnostic laboratories, and “Specific disinfection procedures” for the experimental animal rooms). In addition, for each section, additional new and/or updated requirements were described. Therefore, Appendix A of Commission Decision 2003/422/EC [18] reporting the requirements for ASF laboratories and animal facility were modified as reported in Table 1 and Table 2.

### 3.3. Application of Internal Audit Activity at the NRL for ASF

The internal audit occurred annually to ensure the proper application and the efficacy of the biosafety/biosecurity system and avoid every accidental or intentional release of ASFV infected material outside the BSL-3 containment area. Considering the complexity of the checklist, an internal audit was performed each time with different queries never investigated before. This approach allowed for every aspect of the biosafety/biosecurity system to be checked, gradually. The audit report included findings and observations. The audit findings highlighted significant security holes that required immediate intervention, while observations included suggestions aimed at improving the biosafety/biosecurity system further. As a demonstration, we report some significant audit findings and one observation that arose over time during the internal audit process in Table 3.

## 4. Discussion

Health biotechnology and bioengineering have recently undergone major advances with respect to both human and animal medicine. Since microorganisms may be harmful or pathogenic to animals and humans, to guarantee public health, every possible scenario of an outbreak occurring as a result of the release of a biohazard into the environment cannot be neglected [27]. This implies putting specific biosafety and biosecurity procedures in place. The ever-growing handling of ASFV in laboratories, necessary for either diagnostic or research in this period, requires particular attention to avoid the release of the virus from the laboratories, with consequent detrimental effects on animal health and the pork industry. WHO, FAO, WOAH, national governments, other international agencies, and biosafety associations have published broad biosafety manuals and guidelines in collaboration with expert groups over time. The specific requirements for ASFV laboratories and animal facilities were reported in the Commission Decision of 26 May 2003 n. 422 [18]. However, recently, the Commission Delegated Regulation (EU) 2020/689 of 17 December 2019 [17] repealed the previous Commission Decision, and specific ASF provisions became suddenly unavailable.

Since standards on biosafety and biosecurity may vary according to the laboratory, institution, or even among countries [28,29], a harmonization of criteria is necessary to avoid discrepancies and provide greater protection of the environment and public health. For this purpose, the biosafety group assisted by the NRL drafted specific recommendations for ASF laboratories and animal facilities that aim to bridge the present regulatory gap. The developed specific requirements were based on provisions reported in the repealed Commission Decision 2003/422/EC [18], in the Directive 2000/54/EC of the European Parliament and of the Council still in force [19], and in the biosafety guidelines of international agencies, as well as on the experience of the Italian NRL in the field.

The developed requirements specific for ASF laboratories and animal facilities exhibit the same scheme adopted by the Commission Decision 2003/422/EC [18] to aid consultation by users. However, “Specific disinfection procedures”, “Handling procedures of infected material”, and “Animals” (only for the ASF ABSL-3 experimental animal rooms) were introduced as new sections. In addition, sub-sections were used to further aid the reader during the consultation of the requirements and to emphasize newly proposed issues, in particular related to the correct functioning of the infrastructure.

The information was reported succinctly, and the words “must”, “should”, and “recommended” were utilized in association with mandatory or optional provisions. Standard practices and recommendations necessary to ensure the biosafety of the laboratories, such as bans on eating, drinking, smoking, etc., were not included, unlike other biosafety manuals previously proposed by international agencies. Herein, we report only the structural, equipment, and procedural requirements specific to the handling of ASFV by laboratories and animal facilities. More specifically, instructions on structural characteristics of the working area of both laboratories and animal facilities BSL-3 for ASF were added. In particular, the requirement that the working area be air-locked to allow for disinfection was introduced as mandatory. Indeed, periodic disinfections, useful in the ABSL-3 to decontaminate facility rooms, require a sealed environment. In the “General environment” section, the “Atmospheric pressure” subsection of laboratories BSL-3 for ASF entailing assisted negative pressure ventilation was added under the supplementary requirements as a recommended feature. This characteristic was a mandatory requirement for ASF ABSL-3; in the present version, it was considered an added element of biosafety and, therefore, introduced as a suggestion for BSL-3 laboratories with respect to ASF. In the same subsection, a visual indicator displaying real-time pressure differentials and audible alarms were indicated as optional. These recommendations were also included in relation to the ABSL-3 facilities for ASF. An “ASFV repository” subsection was introduced as a requirement for BSL-3 laboratories for ASF. Indeed, generally, in diagnostic and research laboratories dealing with ASF, virus banking is utilized. This practice requires the implementation of precautions to ensure accurate retrieval, proper storage, and disposal. Therefore, in this subsection, any information regarding the proper storage of the ASFV was outlined for compliance with high biosecurity and biosafety levels. In addition, a specific section dedicated to personnel safety was introduced. It was entitled “Windows inspection and personnel safety” and includes some recommended structural requirements and procedures, such as the inspection of the BSL-3 laboratory and animal facility and the recommendation to require a minimum of two individuals when entering the containment area. This section was introduced in relation to both laboratories and animal facilities BSL-3 dealing with ASF. Moreover, a “Facilities organization” subsection was added to both BSL-3 laboratory and animal facility to provide some standard guidelines useful for the design of laboratory facilities that deal with ASFV. Thus, infrastructure endowment, a proper subdivision of the premises, and service requirements were clearly reported. The section “Equipment”, reporting the requirements for BSL-3 laboratories in relation to ASFV, was separated into three subsections. With respect to “Biological safety cabinet” and “Other equipment”, the original guidelines (2003/422/EC) [18] were maintained. A section entitled “Autoclave” was added to emphasize the necessity to include such instrumentation in ASFV-dedicated facilities. Similarly, with respect to animal facilities, the “Equipment” section was divided into two subsections, one for the safe disposal and the other for all the necessary equipment. Therefore, “Approved means for safe disposal” was introduced to provide guidelines on the proper and safe disposal of large, infected animal carcasses and other type of waste (solid and liquid). More specifically, an incinerator or other approved piece of equipment was suggested. However, to allow for the proper functioning of the animal facility without an incinerator, strict provisions necessary for the safe disposal of waste and carcasses were introduced.

Regarding the “Personnel” section, guidelines on personnel training were introduced in relation to both laboratories and animal facilities, as specific and regular technical and scientific training activities were considered essential. A further novelty concerned the introduction of the “Specific disinfection procedures”. The section was split into two subsections for laboratories and animal facilities: “Potentially or effectively contaminated liquid effluents” and “Potentially or effectively contaminated solid waste”, where specific guidelines were included. Both heat and chemical inactivation treatments for ASFV were mentioned, and specific wastewater treatment was indicated as a requirement. Under the requirements for animal facilities, more specific guidelines were reported for the disposal of solid waste. Indeed, to allow for the safe disposal of solid waste, even from the animal facilities that do not have an incinerator, information on proper storage of waste was added. More specifically, the solid and liquid waste can be stored in containers that must be chemically and externally decontaminated before proper disposal, as per Regulation 1069/2009 [30]. Similarly, under the section “Animals” referring to the requirements for animal facilities, information on proper disposal of carcasses resulting from in vivo experimentation was included also for facilities without incinerators. For a well-functioning biosecurity system, the proper implementation of the requirements must be accompanied by an effective risk assessment and an internal audit system. The risk assessment has to be considered as a dynamic system necessary from the beginning of the BSL-3 containment area activity and throughout its lifetime when structural and process changes occur and new risks might, therefore, be identified.

The developed risk assessment, based on the biological risk analysis process flowchart described in the WOAH Terrestrial Animal Health Manual [24], identifies the level of risk associated with each hazard. In order to reduce each risk to acceptable levels, specific operating and management modalities must be adopted, and specific procedures were, therefore, drawn up. It is not a static process, but rather a constantly evolving one based on a few, simple yet fundamental steps: (1) information gathering, (2) risk evaluation, (3) risk control strategy development, (4) risk control measure selection and implementation, (5) risks and risk control measures review [24].

On the other hand, the internal audit system is a tool to ensure that the biological containment measures, internal procedures, and effective risk assessment are properly implemented, ensuring the best functioning of the biosafety/biosecurity system and avoiding any accidental or intentional release of ASFV-infected material outside the BSL-3 containment area. The internal audit represents an essential tool to highlight critical aspects of the BSL-3 containment area of the Italian NRL in relation to ASFV. All critical audit findings emerged during the first period of the BSL-3 containment area activity. They were related to “Infrastructure management”, a macro-area checking procedural issues, disposal and release of materials, access to the working area, and management of premises and equipment. The ABSL-3 animal facility constitutes a valuable containment premise for the conduction of in vivo trial activities and, therefore, usable also for in vivo studies relying on other biological agents that require a containment level lower than three. However, to ensure a high biosafety level, strict provisions must be indicated and fulfilled when this situation occurs.

The internal audit performed during the first years of activity of the BSL-3 containment area highlighted the absence of a specific procedure for the regulation of the access to and of the activities occurring within the ABSL-3 animal facilities for personnel dedicated to research activities with biological agents different from ASFV. This quality system gap was bridged by making a change to the pre-existing procedures consisting of the addition of specific provisions for activities with biological agents different from ASFV.

An infrastructural gap detected by the internal audit was related to the stream of samples between the laboratory and the animal facility. The laboratory and the animal facility of the BSL-3 containment area must be clearly separated. Despite the existence of a door with a key, the passage from the laboratory to the animal facility and vice versa by named personnel only could not be ensured. Therefore, the installation of an electronic door lock with a code and numeric keypad to regulate the access between the two premises for the BSL-3 containment area was introduced. In addition, a procedural gap was identified. Indeed, a stream of samples can be generated during the experimental trials (samples coming from the animal facility towards the laboratory for diagnostic activities and viruses coming from the laboratory toward the animal facilities for trials). The internal audit showed that no clear provisions were in place for the passage of samples between the laboratory and the animal facilities. Therefore, specific changes to the procedure that regulates access to and activities in the containment area were requested, and instructions on methodologies for the transfer of biological material from animal facilities to the laboratory and vice versa were introduced.

Moreover, the internal audit activity suggested the development of an ASFV repository. A software-based system to archive ASFV biological material, virus strains, and related information was considered necessary and, therefore, introduced.

## 5. Conclusions

Developing high-quality public standard procedures represents the right way to ensure a high biosafety and biosecurity system, improving global public health and protecting the agro-livestock sector. Although aspects of biosafety, biosecurity, and responsible conduct are common in the literature, the lack of integration leaves space for improvement [31]. The requirements for ASF laboratories and animal facilities, the risk assessment, and the internal audit developed by Italian NRL has led to the drafting of a main document (a biosafety and biosecurity manual) with several linked SOPs.

The provisions reported in the present work, alongside the risk assessment and internal audit system resulting from the efforts and substantial experience of the Italian NRL in relation to ASF, is an example of updated and more comprehensive specific guidelines. These may ensure high biosafety and biosecurity standards for laboratories working with ASFV, as well as representing a useful tool for those who have to design laboratory facilities for the handling of ASFV.

Thanks to this complex biological containment system, along with a major shift in mindset, resting on transversal capabilities, a great improvement of the biological containment system was obtained in the Italian NRL ASF BSL-3 containment area.

## Figures and Tables

**Figure 1 animals-14-00454-f001:**
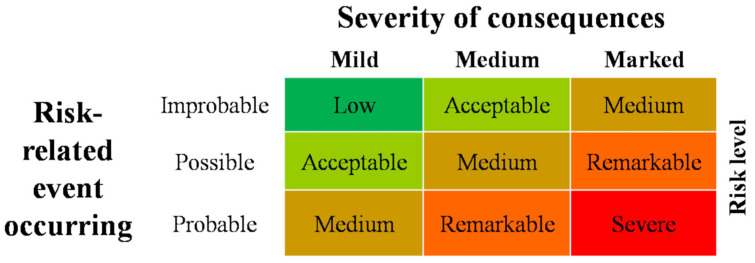
Diagram used to obtain the risk level.

**Table 1 animals-14-00454-t001:** Biosafety requirements for Level 3 (BSL-3) laboratories with respect to ASFV. In bold are sections present in the repealed Commission Decision 2003/422/EC.

Principles of ASFV Biological Containment Appropriate for Diagnostic Laboratories			
Section	Containment Measures	Minimum Requirements	Additional Requirements
**General environment**	Working area	The working area must be separated from any other activity in the same building. Dedicated rooms limited to defined procedures.	The working area must be separated from any other activity in the same building. Exclusively ASFV-dedicated rooms limited to defined procedures.
	Atmospheric pressure	Normal atmospheric pressure.	Assisted negative pressure ventilation is recommended. A visual indicator that displays real-time pressure differentials should be available inside and outside the containment facility to confirm that personnel can enter safely. Audible alarms should be considered to notify personnel of air flow disruption.
	Air filtering system	Vacuum lines must be protected with at least one HEPA ^1^ filter.	
	Working area	The working area must be air-locked to allow for disinfection.	
	Vector control	Efficient vector control (rodents and insects).	
	Surfaces	Surfaces must be water repellent and easy to clean. Carpets and rugs are not permitted. Seams, floors, walls, windows, and ceiling surfaces should be sealed. Spaces around doors and ventilation openings should suitable for sealing to facilitate space decontamination. Surfaces must be resistant to acids, alkalis, solvents, and disinfectants.	
	ASFV repository	All stocks of the ASF virus must be kept in secure storage, whether frozen or freeze-dried. All individual ampoules must be clearly labelled, and comprehensive records maintained of virus stocks together with dates and results of quality control checks. Records must also be kept of viruses added to stock, with details of the source, and of viruses issued to other laboratories.	
	Windows inspection and personnel safety	An inspection window or other device which allows for the unobstructed view of its occupants and for the entry of at least two people in the diagnostic laboratory is recommended.	
	Facilities organization	The facility must be arranged so that personnel ingress and egress are only possible through a series of rooms consisting of: - A ventilated vestibule with compressible gaskets on the two doors. - A “clean” changing room outside the containment area. - A shower room at the non-containment/containment boundary (filter area). - A “dirty” changing room within the containment area. Room within the laboratory for ASFV stock with controlled access. Storage devices must be kept closed and should be accessible to authorized personnel only. Toilets for staff must be available outside the containment barrier in the “clean” changing room. An emergency eye washer and hand-washing basins must be available in the laboratory. A shower must be available in the access filter area. It must be used in case of emergency. A biohazard sign must be present on the door.	
**Equipment**	Biological safety cabinet	Biological safety cabinet (class II) used for all manipulations of live virus. Cabinet should have double HEPA ^1^ filtration of exhaust air.	
	Autoclave	An autoclave must be available in the facility, preferably within the laboratory.	An autoclave must be available within the laboratory.
	Other equipment	All equipment needed for laboratory procedures to be available within the dedicated laboratory suite.	
**Personnel**	Entry	Entry to unit limited to named, trained personnel.	Entry to unit limited to named, trained personnel by clearing house.
	Personal dispositions	Wash and disinfect hands upon leaving the unit. Personnel not permitted to visit premises with pigs for a minimum of 48 h ^2^ after leaving the unit.	
	Personnel training	Laboratory personnel has specific technical and scientific skills. Personnel to receive updates and additional training when equipment, procedures, or policies change. An institutional policy regarding visitor training, occupational health requirements, and safety communication is considered.	
**Laboratory clothing**		Dedicated outer clothing to be used only in the ASF virus unit. Disposable gloves to be used for all manipulations of infected material. Outer clothing to be sterilized before removal from the unit or washed at a high temperature within the unit.	Complete change of clothes upon entry. Laboratory clothing to be used only in the ASF virus unit. Disposable gloves to be used for all manipulations of infected material. Clothing to be sterilized before removal from the unit or washed at a high temperature within the unit.
Specific disinfection procedures	Potentially or effective contaminated liquid effluents.	Liquid waste effluents must be treated to inactivate the ASF virus (heat or chemical). A wastewater treatment is required.	
	Potentially or effective contaminated solid waste.	Solid waste must be treated to inactivate the ASF virus (heat/incineration or chemical).	
Handling procedures of infected material	Infectious material.	Infectious material must be handled in safety cabinets.	
	Inactivated antigens	The processing of tissues for direct immunofluorescence test (DIFT), molecular biology (PCR), or serology using inactivated antigens may be conducted at a lower containment level, provided that the minimum requirements reported in the present table are fulfilled, basic hygiene is maintained, and post-operational disinfection with the safe disposal of carcasses, tissues, and sera is conducted.	

^1^ High Efficiency Particulate Air (HEPA) filter. ^2^ The duration of the ban for personnel to visit premises with pigs is based on the indications reported in the FAO document [26], proposed as a valid minimum requirements for biosafety measures in the containment working area as well.

**Table 2 animals-14-00454-t002:** Biosafety requirements for Level 3 (ABSL-3) experimental animal rooms with respect to ASFV. In bold are sections present in the repealed Commission Decision 2003/422/EC.

Biosafety Requirements for ASF Experimental Animal Rooms		
Section	Containment Measures	Requirements
**General environment**	Working area	The working area must be separated from any other activity in the same building. Exclusively ASFV-dedicated rooms limited to defined procedures.
	Atmospheric pressure	Assisted negative pressure ventilation. A visual indicator that displays real-time pressure differentials should be available inside and outside the containment facility to confirm that personnel can enter safely. Audible alarms should be considered to notify personnel of air flow disruption.
	Air filtering system	Vacuum lines must be protected with at least one HEPA ^1^ filter.
	Working area	The working area must be air-locked to allow for disinfection at end of experiment.
	Vector control	Efficient vector control (rodents and insects).
	Surfaces	Surfaces must be water repellent and easy to clean. Carpets and rugs are not permitted. Seams, floors, walls, windows, and ceiling surfaces should be sealed. Spaces around doors and ventilation openings should be suitable for sealing to facilitate space decontamination. Surfaces must be resistant to acids, alkalis, solvents, and disinfectants.
	Windows inspection and personnel safety	An inspection window or other device which allows for the unobstructed view of its occupants and for the entry of at least two people in the experimental animal room is recommended.
	Facilities organization	The facility must be arranged so that personnel ingress and egress are only possible through a series of rooms consisting of: - A ventilated vestibule with compressible gaskets on the two doors. - A “clean” changing room outside the containment area. - A shower room at the non-containment/containment boundary (filter area). - A “dirty” changing room within the containment area. Animal facilities must be separated by lockable doors. Toilets for staff must be available outside the containment barrier in the “clean” changing room. An emergency eye washer and hand-washing basins must be available in the laboratory. A shower must be available in the access filter area. A biohazard sign must be present on the door.
**Equipment**	Approved mean for safe disposal	Pathological incinerators, or other approved means, are recommended for the safe disposal of the large carcasses of infected animals and other solid waste.
	Other equipment	All equipment needed for laboratory procedures to be available within the dedicated laboratory suite.
**Personnel**	Entry	Entry to unit limited to named personnel by clearing house.
	Personal dispositions	Personnel not permitted to visit premises with pigs for a minimum of 48 h ^2^ after leaving the unit.
	Personnel training	Laboratory personnel has specific technical and scientific skills. Personnel to receive updates and additional training when equipment, procedures, or policies change. An institutional policy regarding visitor training, occupational health requirements, and safety communication is considered.
**Laboratory clothing**	Personnel change on entry	Complete change of clothes upon entry. Complete animal facility clothing (including undergarments, pants and shirts or jump suits, and shoes and gloves) must be provided in the “dirty” changing room or, alternatively, in the “clean” changing room and put on by personnel before entering the research areas.
	Personnel change on exit	Complete change of clothes upon exit. When leaving an ABSL-3 animal space that acts as the primary barrier and that contains large volumes of aerosols containing highly infectious agents (an animal room, necropsy room, carcass disposal area, contaminated corridor, etc.), personnel must: Remove “dirty” lab clothing. Take a shower. Wear “clean” clothing immediately after leaving this high-risk animal space and before going to any other part of the ABSL-3 facility. Soiled clothing worn in an ABSL-3 space must be eliminated or autoclaved before being laundered.
Specific disinfection procedures	Potentially or effective contaminated liquid effluents	Liquid waste effluents must be treated to inactivate the ASF virus (heat or chemical). A wastewater treatment is required.
	Potentially or effective contaminated solid waste	Solid waste must be treated to inactivate the ASF virus (heat/incineration or chemical) or stored in containers that must be chemically and externally decontaminated before proper disposal as per Regulation 1069/2009 of the European Parliament and of the Council.
Animals		All animals must be slaughtered before leaving the unit, and post-mortem examinations are to be completed within the biosafe area. Carcasses must be treated to inactivate the ASF virus (heat/incineration) or cut and stored in containers that must be chemically and externally decontaminated before proper disposal as per Regulation 1069/2009 of the European Parliament and of the Council.

^1^ High Efficiency Particulate Air (HEPA) filter. ^2^ The duration of the ban for personnel to visit premises with pigs is based on the indications reported in the FAO document [26], proposed as valid minimum requirements for biosafety measures in the containment working area as well.

**Table 3 animals-14-00454-t003:** Internal audit results: gaps and actions whose implementation has been necessary for the biosafety and biosecurity of the containment area at the NRL.

Findings			
Activity	Requirements	GAP	Action
Possibility to use ABSL-3 animal facility for trials with biological agents requiring a BSL lower than three when activities with ASFV are suspended.	Secondary independent controlled access to the ABSL-3 animal facility dedicated exclusively to the authorized personnel working with biological agents different from ASFV. SOPs to access the ABSL-3 animal facility for personnel working with biological agents different from ASFV.	Lack of SOPs to access the ABSL-3 animal facility for personnel working with biological agents different from ASFV.	Writing ad hoc SOPs
2. Stream of samples between the laboratory and animal facility.	Physical separation between the animal facilities of the BSL-3 and the laboratory. Passage of authorized, named personnel only from the laboratory to the animal facilities and vice versa. SOPs on sample exchange between BSL-3 diagnostic laboratory and ABSL-3 animal facility.	Lack of lockable doors between animal facilities of the BSL-3 and laboratory. Lack of SOPs on sample exchange between BSL-3 diagnostic laboratory and ABSL-3 animal facility.	Implementation of lockable doors between animal facilities of the BSL-3 and laboratory Writing ad hoc SOPs.
**Observations**			
**Activity**	**Requirements**	**GAP**	**Action**
3. Proper storage and preservation of biological material.	ASF virus repository.	Presence of a virus repository managed by complex, cumbersome, and not completely safe software for the loading/unloading of aliquots.	Implementation of the new updated software-based system for the ASFV repository.

## Data Availability

Data are contained within the article and Appendix A.

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
