# Peer review of "Biological Containment for African Swine Fever (ASF) Laboratories and Animal Facilities: The Italian Challenge in Bridging the Present Regulatory Gap and Enhancing Biosafety and Biosecurity Measures"

_animals, 2024, doi:10.3390/ani14030454_

Round 1

Reviewer 1 Report

Comments and Suggestions for Authors

Comments on the Quality of English Language

Overall, the quality of the authors writing is good and all key concepts are effectively conveyed. There are English language translation errors throughout the manuscript, that should be corrected before publication.  There are several sentence structure issues that I identified in my comments above.

Author Response

Dear Reviewer,

We are glad that the manuscript was appreciated as a useful contribution to biosafety and biosecurity, able to give a guidance document on safe practices when working with AFSV.

We sincerely thank you for helping us significantly improve the manuscript.

We made the changes following the suggestions received to satisfy the requests.

 Summary:

The authors of the paper seek to publish provide a biosafety and biosecurity tool which provides guidance for safely working with African Swine Fever Virus (ASFV) both in vitro and in vivo that can be used globally as a standard upon which local governments and research labs can enhance based on local regulations. This effort is in response to the repeal of the European Union Commission Decision of 26 May 2003 which described a diagnostic manual for ASFV. The authors used this repealed document as the basis for their tool and through an internally developed audit system and risk assessment suggest enhancements to the original document’s recommendations.

General Comments:

The authors seek to fill a gap that exists currently for researchers in the European Union following the repeal of a guidance document on safe practices when working with AFSV. They used the repealed document as the basis for their biosafety and biosecurity tool they did not take the opportunity to evaluate that guidance and correct potential safety issues that could occur when following the guidance. They did offer some enhancements to the original guidance which are well received and complement the core set of guidelines and recommendations. Overall, the presentation of the updated requirements for work with ASFV are presented in an easy-to-read way, but there is a lack of explanation or footnote for some of the requirements that would be helpful to the reader.

R: modified as suggested. A footnote for some requirements reported in the tables has been added as reported below:

  • Table 1.
    • [1] High Efficiency Particulate Air (HEPA) filter
    • [2] The duration of ban to visit premises with pigs by personnel is based on the indications reported in FAO document [26], proposed as a valid minimum requirment for biosafety measures in the containment area working as well.
  • Table 2.
    • [1] High Efficiency Particulate Air (HEPA) filter
    • [2] The duration of ban to visit premises with pigs by personnel is based on the indications reported in FAO document [26], proposed as valid minimum requiremtns for biosafety measures in the containment area working as well.

Additionally, the risk assessment that was conducted would work better if presented before the tables of requirements.

R: modified accordingly. The risk assessment in Materials&Method and Results was shifted before the requirements.

Finally, while it is admirable that the authors revealed the results and lessons learned from their internal audit process, the data might be better presented in a table with only an explanation of the process.

R: as suggested, a table has been added (Table n 3, page 16) in substitution to the text.

The information presented in this manuscript is not wholly from the authors, but an attempt to provide a resource for fellow researchers in need of guidance for working safely with ASFV, with this in mind, I believe this manuscript could be a valuable publication, but it needs revision and reorganization before publication.

Specific Comments:

Below I provide both specific and general comments about the manuscript.

  • On line 45 of the manuscript, delete the “A” at the beginning of the sentence.

         R: modified accordingly.

  • On line 52-32, the definition of biosafety attributed to WHO, is inaccurate. I recommend using the exact quote found on the WHO website (https://www.emro.who.int/health-topics/biosafety/index.html).

R: The suggestion has been received.

  • On line 79, replace “contrastingly” with “similarly”.

R: The suggestion has been received.

  • The section from lines 96-101 describing the risk groups attributed to ASFV is incorrect and I could not verify the information at the cited sources. While I understand why the authors might want to include this type of information, it is extraneous in the context of what they want to convey and I recommend deleting these lines and starting the paragraph from the sentence that starts “Based on a risk assessment…” on line 102.

R: as suggested by the referee lines 96-101 were removed.

  • Following line 104, it would be important to clarify what type of animal work is conducted at ABSL-3 by specifically mentioning the species to be used. The containment requirements and risk of working with mice infected with ASFV is very different than those for working with pigs when talking about animal facilities.

R: as suggested the description of the animal work conducted has been added in the following way: “…for swine used in vaccine efficacy and pathogenicity studies, as well as production of reference material.”.

  • On line 184-187, these sentences should be rewritten as its structure represents bad English grammar. “The risk assessment is an essential process to be completed prior to beginning laboratory activities and is necessary each time new procedures and/or risks are identified or changes to the facility, equipment or agents occur. The biosafety officer conducts the assessment annually and whenever deemed appropriate and should identify risk factors and potential sources of contamination that are linked to the identified hazard.”

R: corrected. The sentence in lines 184-187 was replaced with that suggested by referee.

  • Line 191, change “envisaged” to “implemented.”

R: The suggestion has been received.

  • Line 192-194, rewrite sentence. “When risk is unable to be mitigated to an acceptable level, the biosafety officer communicates the hazard to the facility director, who is able to order a suspension of work activities involved with the process with an unacceptable risk factor.”

R: corrected. The sentence in lines 192-194 was re-phrased according to the suggestion.

  • Lin 219, move “ASFV” before the words “infected material”.

R: The suggestion has been received.

  • In Table 1, Facilities organization, change “storages must be kept closed…” to “storage devices must be kept closed…”

R: The suggestion has been received.

  • In Table 1, Facilities organization, I recommend clarifying where the toilet should be located in the facility. I do not recommend locating a toilet within the containment area as the risk of self-contamination is much higher than it would be if the toilet is located outside the containment barrier in the “clean change” room.

R: The suggestion has been received and the change was made also in Table 2.

  • In Table 1, Biological Safety Cabinet, I recommend removing Class I BSCs as an option as they do not provide any product protection and are inferior to Class II biological safety cabinets for providing protection to workers, samples and the environment simultaneously.

R:The suggestion has been received.

  • In Table 1, Personnel, recommend changing quarantine time after contact with virus or infected animals to 4 days from 48 hours based on data presented in Food & Agriculture Organization of The United Nations, “Preparation of African Swine Fever Contingency Plans”, Chapter 2, 2009.

                R: Thank you for your suggestion. In the “Preparation of African Swine Fever Contingency Plans” (2009), on page 45,  the following sentence “Compartments should be inspected by individuals or teams that have not visited infected premises for at least 72 hours beforehand, in case there has been a breach of contamination procedures.” is reported. In addition, a more recent document entitled “African swine fever in wild boar Ecology and biosecurity” (FAO Animal Production And Health, 2022, Second edition) on page 71, reports “The personnel responsible for carcass disposal or transport must be trained on ASF and biosecurity.[…] Involved personnel must not have any direct contact with healthy pigs for 48 hours”. https://www.woah.org/app/uploads/2022/07/asf-in-wild-boar-ecology-and-biosecurity-2nd-ed.pdf

We therefore re-phrased the sentence adding minimum 48 hours

  • Line 309, “The risk level or of each hazard, expect except hazard n.7….”

R: The suggestion has been received.

  • Line 378, change “ensure” to “be achieved”.

R: The suggestion has been received.

  • Line 445, change “…to enter in the containment area at least two persons.” To “…to require a minimum of two persons when entering the containment area.”

R:The suggestion has been received.

  • In the References section, reference 10 and 15 appear to be the same source. Eliminate one and adjust references throughout the manuscript

                R:The duplicated reference has been removed.

Reviewer 2 Report

Comments and Suggestions for Authors

The manuscript titled “Biological containment for African Swine Fever ( ASF ) laboratories and animal facilities : the Italian challenge in bridging the present regulatory gap and enhancing biosafety and biosecurity measures” describes laboratory recommendation from the Italian NRL when working with ASFV in high containment labs. Overall, the manuscript is well- detailed and thorough. The recommendations listed are appropriate to support safe research with ASFV. I have only minor comments to offer:

Line 154: Do the authors mean infected instead of “infeted”?

Table 1: What is the justification for only 48 hours before personnel can visit premised with pigs?

Line 326: Add space between “onpersonnel”

Line 331: Add space between “;SOP”

Author Response

The manuscript titled “Biological containment for African Swine Fever ( ASF ) laboratories and animal facilities : the Italian challenge in bridging the present regulatory gap and enhancing biosafety and biosecurity measures” describes laboratory recommendation from the Italian NRL when working with ASFV in high containment labs. Overall, the manuscript is well- detailed and thorough. The recommendations listed are appropriate to support safe research with ASFV. I have only minor comments to offer:

 We are glad that the manuscript was appreciated as a useful contribution to the biosafety and biosecurity for ASFV. We thank the Reviewer for the comments provided in order to improve the paper. We made the changes following the suggestions received to satisfy the requests.

Line 154: Do the authors mean infected instead of “infeted”?

R: modified as suggested.

Table 1: What is the justification for only 48 hours before personnel can visit premised with pigs?

R: 48 hours represent the time suggested in the most recent version the document “African swine fever in wild boar Ecology and biosecurity” (FAO, 2022, Second edition) avaliable on the following web page: https://www.woah.org/app/uploads/2022/07/asf-in-wild-boar-ecology-and-biosecurity-2nd-ed.pdf In particular, on page 71 is reported, “The personnel responsible for carcass disposal or transport must be trained on ASF and biosecurity.[…] Involved personnel must not have any direct contact with healthy pigs for 48 hours”.

We considered this indication valid also for biosafety measures in the containment area working. We add this information as a footnote in the tables.

Line 326: Add space between “onpersonnel”

R: modified accordingly

Line 331: Add space between “;SOP”

R: modified accordingly.

Round 2

Reviewer 1 Report

Comments and Suggestions for Authors

I commend the authors for accepting the suggested edits.  I feel the paper is much stronger now then in the first iteration.  While I still feel that the quarantine period following handling of agent should be more than 48 hours, the addition of "minimum" to the guidance is acceptable.